# Wnt/β-Catenin Signaling as a Driver of Stemness and Metabolic Reprogramming in Hepatocellular Carcinoma

**DOI:** 10.3390/cancers14215468

**Published:** 2022-11-07

**Authors:** Rainbow Wing Hei Leung, Terence Kin Wah Lee

**Affiliations:** 1Department of Applied Biology and Chemical Technology, The Hong Kong Polytechnic University, Hong Kong, China; 2State Key Laboratory of Chemical Biology and Drug Discovery, The Hong Kong Polytechnic University, Hong Kong, China

**Keywords:** cancer metabolism, drug resistance, metabolic reprogramming, hepatocellular carcinoma, Wnt/β-catenin

## Abstract

**Simple Summary:**

Aberrant Wnt/β-catenin signaling has been reported to play crucial role in pathogenesis of hepatocellular carcinoma (HCC). In this review, we focus on the regulatory role of Wnt/β-catenin signaling in cancer stemness and metabolic reprogramming, which are two emerging hallmarks of cancer. Understanding the role of Wnt/β-catenin signaling in regulation of the above processes reveals novel therapeutic strategy against this deadly disease.

**Abstract:**

Hepatocellular carcinoma (HCC) is a major cause of cancer death worldwide due to its high rates of tumor recurrence and metastasis. Aberrant Wnt/β-catenin signaling has been shown to play a significant role in HCC development, progression and clinical impact on tumor behavior. Accumulating evidence has revealed the critical involvement of Wnt/β-catenin signaling in driving cancer stemness and metabolic reprogramming, which are regarded as emerging cancer hallmarks. In this review, we summarize the regulatory mechanism of Wnt/β-catenin signaling and its role in HCC. Furthermore, we provide an update on the regulatory roles of Wnt/β-catenin signaling in metabolic reprogramming, cancer stemness and drug resistance in HCC. We also provide an update on preclinical and clinical studies targeting Wnt/β-catenin signaling alone or in combination with current therapies for effective cancer therapy. This review provides insights into the current opportunities and challenges of targeting this signaling pathway in HCC.

## 1. Introduction of Hepatocellular Carcinoma (HCC)

Liver cancer was the third leading cause of death from cancer worldwide in 2020 according to the World Health Organization, where hepatocellular carcinoma (HCC) is the dominant primary liver cancer, accounting for over 90% of cases [1]. HCC is often found in patients with liver cirrhosis, with higher incidence and mortality rates in men as compared to women [1,2]. The major risk factors include chronic viral hepatitis (hepatitis B virus and hepatitis C virus) infection, excessive consumption of alcohol, and exposure to chemicals such as aflatoxin [3]. Several studies have reported that metabolic syndrome with accumulating fats in the liver causes nonalcoholic steatohepatitis (NASH) and can also lead to advanced fibrosis and cirrhosis [4]. Nonalcoholic fatty liver disease (NAFLD)-induced HCC is an emerging malignancy in developed countries.

With the continuously increasing trend of HCC cases, there are several therapeutic strategies ranging from curative therapy and targeted drug therapy to immunotherapy. Liver transplantation, resection or ablation are suggested to be the suitable therapeutic modalities for HCC patients at an early stage without intrahepatic or extrahepatic metastases [5]. There is a variety of drugs approved by the US Food and Drug Administration (FDA) specifically targeted for HCC patients in advanced stages. The multityrosine kinase inhibitors (TKI) sorafenib and lenvatinib are commonly utilized as first-line molecular targeted drugs orally administered. Meanwhile, regorafenib, cabozantinib, and ramucirumab are used as second-line treatments to prolong patient survival, yet their therapeutic efficacies remain unsatisfactory, possibly due to the development of drug resistance [6]. Nivolumab and pembrolizumab have been available as single-agent second-line therapies for patients with HCC, but their response rates are less than 20% [7]. Recently, bevacizumab in combination with atezolizumab showed superior therapeutic efficacy over sorafenib treatment [8]. Thus, understanding the molecular mechanisms in regulation of therapeutic resistance are urgently needed for novel combinatorial strategies for improvement of the prognosis of HCC patients. 

## 2. Canonical and Non-Canonical Pathways of Wnt Signaling

The Wnt signaling pathway partakes in various essential biological processes, including embryonic development, maintaining adult stem cells, and regulating proliferation and angiogenesis. It is classified into canonical and noncanonical pathways, where the former is β-catenin dependent, whereas the latter is β-catenin independent [9]. In the canonical Wnt pathway, several key intracellular signal transducers have been identified, including a scaffolding protein named dishevelled molecule (Dvl/Dsh), casein kinase 1 alpha (CK1α), glycogen synthase kinase-3 (GSK3β), Axin1 and adenomatous polyposis coli protein (APC), in which β-catenin is the main component [10]. There are two states of the canonical Wnt signaling pathway, depending on the presence of β-catenin. In the “Wnt-off” state, β-catenin is degraded by phosphorylation in the degradation complex consisting of CK1α, GSK3β, Axin and APC in the absence of Wnt ligand bound to Frizzled receptor (FZD) and low-density lipoprotein receptor-related protein (LRP) 5 and 6 via the ubiquitin proteasome system (UPS). The level of β-catenin in the cytoplasm decreases and is unable to drive the transcription of Wnt target genes. Conversely, when the Wnt ligand is bound, the destruction complex is cleaved, and Axin is recruited to the LRP5/6 receptor, leading to the “Wnt-on” state. β-catenin is unable to be phosphorylated due to the disassembled destruction complex. As a result, β-catenin escapes degradation, enters the nucleus and binds to the T-cell factor/lymphoid enhancer factor (TCF/TEF) family members, driving the expression of Wnt target genes [10,11].

The noncanonical signaling pathway is classified into the Wnt-planar cell polarity pathway (PCP) and the Wnt/calcium pathway. Both are also activated by the Wnt ligand attached to the FZD receptor but play different roles in regulating the cytoskeletons and calcium ions within the cells [12]. In the PCP pathway, Dvl/Dsh is recruited to FZD and induces the small GTPases Rho and Rac in parallel pathways. Rho-mediated signaling activates Rho-associated kinase (ROCK) to regulate cytoskeletal dynamics, while Rac stimulates JNK via MAPK signaling pathway activation for cell polymerization [13]. The Wnt/calcium pathway is firstly activated by G proteins with the recruitment of Dvl/Dsh to FZD, leading to the activation of phospholipase C (PLC). Phosphatidylinositol 4,5-bisphosphate (PIP2) is subsequently cleaved into diacylglycerol (DAG) and inositol 1,4,5-trisphosphate (IP3), which further induces the phosphorylation of PKCs and thus increases cytosolic calcium ions. Calcium flux promotes the activation of Calcineurin and calmodulin-dependent protein kinase II (CAMKII), leading to the activation of nuclear factor of activated T cells (NFAT) to drive calcium-dependent target genes, while nemo-like kinase (NLK) is activated by CAMKII, which functions as a β-catenin-TCF inhibitor to hinder the transcription of Wnt target genes [13,14,15]. Hence, this reveals the crosstalk between the canonical Wnt pathway and the Wnt/calcium pathway.

## 3. The Role of Wnt/β-Catenin Signaling Pathway in HCC

### 3.1. Mutation and Expression Status of the Wnt/β-Catenin Pathway and Its Clinical Significance

Wnt/β-catenin signaling is crucial in contributing to HCC pathogenesis, where genetic mutations and epigenetic alterations are primarily revealed [10]. Activation of the Wnt/β-catenin signaling pathway was discovered in 20–35% of HCC cases, among which most are resulted by gene mutations of the key genes, including *CTNNB1*, *AXIN*, and *APC* [16,17,18]. In this case, *CTNNB1* is the gene that specifically encodes β-catenin. Mutation of β-catenin is positively related to HCC progression due to its oncogenic role [19]. To date, mutations at the serine/threonine sites of exon 3 of the β-catenin gene are mostly found to be involved in the phosphorylation and ubiquitination of β-catenin, thus enhancing its nuclear translocation in approximately 20% of HCC cases [20,21]. In addition, conventional and missense mutations have also been reported in other codons of β-catenin [22]. Previous reports showed that conventional mutations at codons 33, 37, 41, and 45 are discovered in over 12% of HCC patients, where missense mutations are observed at codons 32, 34, and 35 [22], which indicates the capability of mutated β-catenin proteins to evade degradation and enter the nucleus [20,23]. It is also noted that tumor cells with aberrant Wnt/β-catenin activation due to the mutation of β-catenin that tend to grow and spread more quickly in HCC [19].

Apart from β-catenin, deregulation of the Wnt/β-catenin signaling pathway is also caused by mutations in protein degradation complexes [24]. These mutations cause dysfunction of the destruction complex and accumulation of β-catenin in the nucleus in approximately 40–70% of HCC cases [10]. One example is the amino acid substitution in armadillo repeats domain 5/6 of β-catenin in human HCC cases [25]. This results in a reduction of APC binding to the degradation complex, which activates the Wnt/β-catenin signaling pathway and enhances targeted gene transcription [25]. It has been reported that a small amount of β-catenin accumulated in the nucleus is sufficient to activate Wnt target genes, suggesting the crucial role of β-catenin in HCC progression [26,27]. However, several studies have shown that the mutation of β-catenin alone is insufficient for promoting HCC in mice, which is different in comparison with humans [27,28], as the tumorigenic potential could be augmented when combined with other oncogenic pathways, such as H-RAS, MET, AKT, or chemicals such as diethylnitrosamine (DEN) [10,29]. 

In addition, high levels of E-cadherin have been reported to be correlated with the accumulation of β-catenin in both the cytosol and nucleus, that drives the transcription of Wnt target genes [9]. *C-Myc* and *cyclin D*, as key Wnt-target genes, not only perform their roles as proto-oncogenes for tumor formation but also regulate liver cancer stem cell (CSC) properties by mediating various signaling pathways involved in cellular differentiation and survival [19,20]. As previously mentioned, HBV and HCV are the causes of HCC, in which they lead to genetic mutations in genes involved in Wnt/β-catenin signaling [9,30]. It is common to find *CTNNB1* mutations in HCV-related HCC rather than HBV-related HCC or nonviral HCC [24]. However, mutation of *Axin1* is more often found in HBV-related HCC tumors [24]. Interestingly, apart from the mutations of the canonical pathway, Zucman-Rossi et al. suggested that *Axin1* mutation also plays a role in exerting oncogenic effects manifested by overexpressing glutamine synthase (GS), leading to β-catenin activation that correlates to the non-canonical pathway [31]. 

### 3.2. Regulation of Wnt/β-Catenin Pathway in HCC

#### 3.2.1. Epigenetic Regulation of Wnt/β-Catenin

Several epigenetic dysregulations contribute to Wnt/β-catenin activation in HCC. DNA methylation is crucial in maintaining CSC properties, in which its inhibition can influence the fate of cells and gene expressions [32]. For instance, DNA methyltransferase (DNMT) plays a role in catalyzing the transition between a methyl group and DNA, mediating BEX1 expression in HCC [33]. A decrease in DNMT1 results in BEX1 hypomethylation that further enhances the transcription of β-catenin, which causes the activation of the Wnt/β-catenin signaling pathway [33]. Moreover, secreted frizzled-related proteins (SFRPs) negatively regulate Wnt/β-catenin signaling via DNA methylation, representing a leading cause of activating β-catenin activity in HCC [34]. Another study also consistently showed that downregulation of the SFRP family is correlated with Wnt/β-catenin signaling activation, in which SFRP1 and SFRP5 are also found to enhance the progression of HCC [35,36]. Similarly, downregulating Wnt inhibitory factor 1 (WIF1) or Dickkopf-related protein 3 (DKK3) has been proven to result in common consequences for SFRPs [37]. In addition, SOX17 is reported to take part in the aberrant activation of Wnt/β-catenin signaling due to promoter methylation [38]. Silencing of SOX17 could enhance Wnt activity due to the failure in interacting with TCF/LEF, which hinders Wnt-target gene transcription [39]. Apart from the genomic instability caused by DNA hypomethylation, another study showed the involvement of potassium channels in epigenetic regulation of the Wnt/β-catenin pathway [40]. Fan et al. revealed that a decrease in KCNQ1 (potassium voltage-gated channel subfamily Q member 1) causes an increment in Wnt/β-catenin activity via DNA hypermethylation [40].

Furthermore, several alterations through histone modification have been reported in HCC. Enhancer of zeste homologous 2 (EZH2) is a histone methyltransferase that plays a role in catalyzing methylation of histone H3 to achieve repression of Wnt antagonists, promoting Wnt/β-catenin signaling and hepatocarcinogenesis [41,42]. Histone deacetylases (HDAC) have been revealed to interact with EZH2 through its enzymatic role [43]. Specifically, for HDAC8, its upregulation due to the chromatin modifications is coexpressed with the lipogenic transcription factor SREBP1 in HCC mouse models, causing cell cycle arrest and β-catenin activation, which drives NAFLD-induced hepatocarcinogenesis [43]. Moreover, HDAC8 can also bind to pyruvate kinase M2 (PKM2) and subsequently deacetylate the residue K62, prompting the nuclear translocation of PKM2 and the binding of β-catenin that results in Wnt target gene transcription [44]. Similarly, EZH2 overexpression elevated the levels of the oncogene H3K27me3, which silenced Wnt inhibitors, leading to induced cell proliferation with activated β-catenin activity [41].

#### 3.2.2. Non-Coding RNAs in Regulation of Wnt/β-Catenin

It has been suggested that microRNAs (miRNAs) and long noncoding RNAs (lncRNAs) are critical regulators associated with various tumors, in which they are negatively regulated [45]. Dysregulation of miRNAs and lncRNAs could lead to tumorigenesis in HCC. LncRNA-miRNA binding yields a complete endogenous RNA (ceRNA) that can avoid messenger RNA (mRNA) recognition and further silencing effects, known as the “sponge effect” [45]. Mounting evidence suggests that miRNA sponges are involved in Wnt/β-catenin signaling and are associated with HCC progression (Table 1). For example, LINC00355 and LINC01278 are negative regulators of miR-217-5p and miR-1258, respectively [46,47]. Overexpression of lncRNAs downregulate the corresponding miRNAs and further activates Wnt/β-catenin signaling, resulting in increased levels of Wnt target gene transcription and metastatic ability of HCC cells [46,47]. Additionally, upregulation of LINC00662 in HCC induced *WNT3A* secretion with miR-15a/16/107 binding, resulting in the activation of Wnt/β-catenin and polarizes M2 macrophage [48]. Similarly, overexpressing FEZF1-AS1 negatively regulates the level of miR-107, which inhibits the activation of Wnt/β-catenin signaling, while downregulation of FEZF1-AS1 enhances the expression of β-catenin [49]. Furthermore, both miR-122 and miR-148a were found to contribute to liver cancers by binding to the 3′-untranslated region (3′-UTR) site of *Wnt1*, suppressing the level of β-catenin and inhibiting Wnt-target gene transcription [50,51]. Furthermore, a decrease in these miRNA levels could cause excess Wnt/β-catenin signaling and increase EMT [50]. All the above mentioned enhance the progression of HCC. As a tumor suppressor, miR-34a is reported in mice and HCC patients and found to be upregulated through activated Wnt/β-catenin signaling [52]. In addition, overexpression of miR-145 has been shown to diminish the level of β-catenin, suppressing HCC cell growth [53]. To sum up, Wnt/β-catenin signaling is tightly regulated by DNA methylation, histone modification and non-coding RNAs in HCC (Figure 1). 

#### 3.2.3. Other Molecules Involved in the Regulation of Wnt/β-Catenin

Apart from genetic mutations and epigenetic dysregulation, other molecules/pathways were identified to regulate Wnt/β-catenin signaling. In normoric environment, ROS is maintained at a low level; whereas a steady increase of ROS level promotes cancer development and progression [70]. A recent study showed that Wnt/β-catenin signaling was suppressed upon elevation of intracellular ROS level [19,71]. In HCC, glutaminase 1 (GLS1) is upregulated which augmented liver CSC properties with increased expression of CSC markers via suppression of ROS level [19,71]. Likewise, another study also showed that ROS accumulation due to the overexpression of Cytochrome P450 2E1 (CYP2E1) decreased the activity of Wnt/β-catenin signaling through the degradation of DVL2 in HCC [72]. Hypoxia also plays a crucial role in the activation of Wnt/β-catenin signaling. Hypoxia-inducible factor 1-alpha (HIF1α), a hypoxia-inducible factor, regulates transcription in hypoxic environments and is also reported to mediate the expression of B-cell lymphoma 9 (BCL9) [9,73]. BCL9 can coactivate with HIF1α to enhance the transcriptional activity of β-catenin regardless of whether genetic mutations occur, resulting in activation of Wnt/β-catenin signaling and leading to HCC progression [73]. Furthermore, *ZBTB20* has been reported in liver tumorigenesis with its role in suppressing PPARG expression and inhibiting proteasomal degradation of the β-catenin destruction complex [74]. Overall, once the nuclear translocation of β-catenin is achieved, the expression levels of the downstream genes involved in EMT are modulated and enhanced, causing hepatocarcinogenesis [17]. *C-Myc* is the most critical gene induced by activated Wnt/β-catenin signaling, which enhances the mechanisms of glycolysis and glutaminolysis [75]. This is followed by cyclin D1, which has been reported to be enhanced in both mouse and human HCC [76,77]. Specifically, overexpression of c-Met and cyclin D1 triggers the development of liver tumors and decreases survival in mice [78]. It is also noted that upregulation of cyclin D1 enhances tumor metastatic ability [79]. Additionally, studies have discovered that GS and VEGF are also involved in modulating the downstream effects of activated Wnt and assisting in angiogenesis [80], as the upregulation of multiple matrix metalloproteinases (MMPs), including MMP2 and MMP9, is associated with tumor metastasis [81]. Apart from gene regulation, aberrant β-catenin signaling also negatively regulates certain signaling cascades: for example, the suppression of NF-κB cascade in the liver [82]. Moreover, the crosstalk between Wnt and Hippo signaling pathways has been observed in HCC. Recent study showed that Wnt-Hippo signature related genes may be a potential markers for prediction of immune infiltration in HCC [83]. Notably, aberrant activation of β-catenin caused by the deletion of mammalian STE20-like protein kinase 1/2 (Mst1/2) promotes tumor growth, indicating the co-expression of YAP and β-catenin in HCC [84].

## 4. Wnt/β-Catenin Signaling in Cancer Metabolism

Altered metabolism, one of the hallmarks of cancer, was recently reported to be linked with Wnt signaling and cancer immunology. When compared with normal cells, cancer cells require more energy and glycolytic intermediates, such as glucose-6-phosphate (G6P) and NADPH, and TCA intermediates, including citrate, acetyl-CoA and lactate, for the biosynthesis of macromolecules to sustain cell proliferation [85]. These intermediates contribute to several metabolic pathways that can influence drug efficacy and lead to cancer aggressiveness [86]. Accumulating evidence supports the role of Wnt/β-catenin signaling in aerobic glycolysis, glutaminolysis and fatty acid metabolism (Figure 2).

### 4.1. The Role of Wnt/β-Catenin in Aerobic Glycolysis

HCC cells are different from normal hepatocytes, as they express various metabolic enzymes [87]. For instance, higher glucose uptake via glucose transporter 1 (GLUT1) and sodium-glucose cotransporter 2 (SGLT2) is required for cancer cells to accomplish conversion to pyruvate and then metabolize pyruvate to lactate, which acidifies the microenvironment and further favors the polarization of tumor-associated macrophages (TAMs), enhancing the migration and invasive abilities of tumor cells [88,89]. GLUT1 and SGLT2 are upregulated in HCC to transport glucose into the cytoplasmic region, with several enzymes and substrates in aerobic glycolysis, the pentose phosphate pathway, gluconeogenesis, and the TCA cycle promoting HCC progression. Yu et al. showed that Wnt/β-catenin signaling initiated by Wnt2b activates glycolysis in HCC-TAMs and promotes macrophage polarization, leading to EMT development and HCC progression [90]. The team also observed a reverse trend when silencing either Wnt2b or β-catenin both in vitro and in vivo [90]. In normal cells, energy is generated in the form of adenosine triphosphate (ATP) via oxidative phosphorylation (OXPHOS), as there is a switch to intense glycolysis to meet greater demand for energy in tumor cells, even when oxygen is sufficient, known as the “Warburg Effect” [91]. Studies have reported that some enzymes change their original form to a specific isoform and are highly expressed in HCC, such as HK2 and PKM2 [92,93]. NADPH and pentose are the products produced from PPP by the G6P isomerase diverging from glucose flow, in which NADPH plays a role in transducing energy to fuel the antioxidant system and fight against oxidative stress [94]. G6P isomerase also serves in suppressing cell death and promoting cancer cell growth via PI3K/AKT activation [95], which can further initiate HCC progression. Furthermore, it has been reported that activated Wnt/β-catenin signaling enhances glycolytic activity through the conversion of pyruvate to oxaloacetate by pyruvate carboxylase, making cancer cells more aggressive [96,97]. Conversely, the disruption of Wnt/β-catenin signaling could downregulate the aerobic glycolysis [98]. Lactate, not only as the carbon source in membrane lipids, but also relates to oxidative stress resistance and lipid biosynthesis in HCC, is the main metabolic product of glycolysis. Lactate can cause ferroptosis resistance in HCC cells regulated by HCAR1/MCT1 [99]. Consistently, autophagy in aberrant Wnt signaling also increases glycolysis and metastatic ability in HCC cells via upregulated MCT1 expression, whereas the decreased accumulation of β-catenin hampers autophagy-induced glycolysis [100]. Lactate provides energy in the TCA cycle in cancer cells [101]. The two processes of cataplerosis and anaplerosis assist in transporting intermediates of the TCA cycle in and out of the mitochondria to stabilize the cycle [102]. For instance, pyruvate carboxylase is an anaplerotic enzyme catalyzing the carboxylation of pyruvate and providing oxaloacetate to support anaplerosis. Thus, the intermediate flux could be maintained for the proliferation of HCC cells [31].

### 4.2. The Role of Wnt/β-Catenin in Glutaminolysis

The demand for glutamine, a free amino acid abundantly present intracellularly, is increased in cancer cells in order to satisfy their metabolic needs compared to normal cells. Glutamine is also involved in redox homeostasis mediated by glutamine transporters, mainly the SoLuteCarrier (SLC) families, which are highly expressed in the liver and serve as metabolic gateways to modulate the transport of both nutrients and metabolites, such as glucose, amino acids, vitamins and certain ions [103,104]. This increase in catabolism is also known as glutaminolysis. Highly proliferating cells are dependent on the conversion of glutamine to glutamate via mitochondrial glutaminase (GLS) and subsequent conversion to α-ketoglutarate (α-KG) through deamination by glutamate dehydrogenase (GLUD), leading glutamate to enter the TCA cycle for high-efficiency ATP production [105]. Studies have shown that overexpressing GLS1 enhances the proliferation of HCC cells, which involves the AKT/GSK3 β/cyclin D1 axis [104], while the identity of CSCs is regulated by GLS1 via the ROS/WNT/β-catenin pathway [71]. Moreover, a rise in glutamine levels results in a high synthesis rate of glutathione (GSH), which reduces the level of ROS and thus drives drug resistance [106]. While another study showed that decreasing GSH levels lead to a rise in ROS level, enhancing β-catenin phosphorylation under the deprivation of glutamine, which further downregulates the Wnt signaling [107]. Furthermore, glutamine synthetase (GS) is a transcriptional target of β-catenin, which has been reported as a reliable marker for identifying CTNNB1-mutated HCC growth [108]. Adeola et al. revealed that *mTORC1* activation is caused by GS-mediated glutamine synthesis, correlated with upregulated β-catenin signaling and phosphorylation of mTOR-S2448 in human HCC, thus revealing the relationship in the GS/glutamine/p-mTOR-S2448 axis [109]. Another study also proved that the expression levels of GS and *ALDH3A1* are positively correlated, in which Wnt/β-catenin signaling is consistently upregulated in HCC [110]. Branched-chain amino acids (BCAAs) and the related supplementation have been widely studied in HCC. Branched Chain Amino Acid Transaminase (BCAT) serves as an enzyme in the metabolism of BCAAs and is proven to modulate the autophagy-related gene expression via activation of PI3K/Akt/mTOR signaling cascade [111]. Another study also showed the role of BCAT1 in promotion of Wnt/β-catenin signaling via upregulation of Wnt-target genes including *c-myc*, *CCND1* and *MMP7*, leading to HCC progression [112]. Furthermore, a study shown that BCAT1 upregulation suppressed cisplatin-induced apoptosis, while inhibiting the expression of BCAT1 sensitised the effect to cisplatin via the blockade of the autophagy response of HCC cells [113]. Additionally, BCAA is reported to contribute in TGF-β1-induced lipogenesis resulting in activation of Wnt/β-catenin signaling in hepatocytes. The levels of Wnt-target genes in HCC cells are enhanced by TGF-β1 in the absence of BCAA, while their expression was highly suppressed with the presence of BCAA [114,115]. 

### 4.3. The Role of Wnt/β-Catenin in Fatty Acid Metabolism

Several components of the fatty acid transportation machinery, such as CD36, are upregulated in malignant tissues compared with normoxic cells and tissues. They are associated with the enhancement of EMT through the activation of WNT/TGF-β signaling [116]. Therefore, fatty acids can be actively transported across the cell membrane. In this situation, the development of HCC is activated by the aberrant formation of new fats. Cancer cells consume cytoplasmic acetyl-CoA as a substrate to synthesize fatty acids. Citrate in the TCA cycle is transported from the mitochondria to the cytoplasm; where, in the presence of a critical enzyme, ATP-citrate lyase (ACLY), citrate is converted into acetyl-CoA and oxaloacetate [117]. ACLY removes citrate in the cytosol to enhance glycolysis, increasing the activity of phosphofructokinase 1 (PFK1) and PFK2 [117]. Then, de novo lipogenesis (DNL) is activated as a consequence of upregulated glycolysis and glutaminolysis, in which acetyl-CoA carboxylase (ACC) is another crucial molecule and has been revealed as an important part of HCC cell proliferation [118,119]. Recently, ACLY in DNL has been reported to modulate Wnt/β-catenin signaling by disrupting the stability of β-catenin in HCC [119]. High expression of ACLY is found in HCC tissues, which correlates with poor survival in HCC patients, while downregulation of ACLY could suppress the CSC properties of HCC cells [119]. In the process of DNL, citrate is first converted to acetyl-CoA, which is further carboxylated to malonyl-CoA by ACC, followed by conversion of unsaturated fatty acids (UFAs) from saturated fatty acids (SFAs) via upregulation of stearoyl-CoA desaturases (SCD1) and fatty acid synthase (FASN) [120,121]. Phosphorylated ACC can lower the expression of malonyl-CoA in the liver to suppress fatty acid synthesis, which has been reported to alleviate the early stages of NAFLD and liver fibrosis [122]. The condensed malonyl-CoA and acetyl-CoA trigger FASN to form other fatty acid synthesis products. Elevated FASN is an indicator of poor prognosis for HCC patients [122]. Some studies have shown that *SCD1* is associated with the modulation of *p53*, Wnt/β-catenin and autophagy [123,124,125]. The oncogene *MYC* has also been shown to be a transcriptional amplifier for certain PPARα target genes, which consequently potentiate to HCC progression [126]. DNL is also closely related to tyrosine kinase inhibitor (TKI) resistance, and there is a positive correlation between lipid droplet density and TKI resistance [118]. Supporting evidence has shown that the accumulation of lipid droplets (LDs) by the upregulation of FASN sustains the survival of cancer cells via suppression of ROS levels [118]. Yao et al. found that the Wnt-targeted gene *MYC* triggers phospholipid conversion and that UFAs are increased, which is beneficial to cancer cells in terms of cell structure maintenance and energy storage [127,128]. Interestingly, studies have revealed crosstalk between DNL and cholesterol biosynthetic pathways in progressing liver cancer, while it is noted that dysregulation of cholesterol biosynthesis is a proven metabolic event observed in HCC patients [129].

### 4.4. The Roles of Growth Factors and Oncogenic Factors in Regulation of Wnt/β-Catenin-Induced Cellular Metabolism

Wnt/β-catenin signaling regulates cellular metabolism through modifying the activities of several oncogenic factors. Activated c-myc regulates glycolysis, glutaminolysis and lipid synthesis within the TME via enhancing gene expression of *GLUT1*, *HK*, *PKM2* and *SLC1A5*, [130]. In addition to c-myc, AMP-activated protein kinase (AMPK), which serves as an energy sensor, was found to regulate the anabolic and catabolic metabolism pathways by lowering Wnt signaling via modulating the activity of the DVL molecule [131]. Furthermore, studies showed that the activity of mTOR is mediated by amino acids, like glutamine [132]. *mTORC1* repressed GLS expression by enhancing the translation of *c-myc* through S6K, which led to an increase in glutaminolysis [133]. Apart from this, mTOR signaling can also upregulate the expression of both glucose transporters and glycolytic enzymes, like GLUT1 [134]. Another study also showed that Wnt/β-catenin was regulated by TSC2/mTOR-dependent GSK3 suppression [134,135]. 

Insulin receptor substrates (IRS) were induced by WNT3a stimulation which led to Akt activation in HCC [136]. During hepatocarcinogenesis, it has been reported that IRS1 not only promotes “Warburg effect” but also regulates the fatty acid metabolism [136]. Additionally, insulin acts as an inducer in promoting de novo lipogenesis via PI3K/mTORC1 signaling cascade, which in turn activates SCD1-induced Wnt/β-catenin signaling [137]. In addition to IRS, other growth factors including EGF, FGF, TGF and their receptors were also reported to be regulated by Wnt/β-catenin signaling [80,138]. For instance, Wnt ligands can activate EGFR signaling via the 7-transmembrane domain receptor Fizzled while EGFR in turn activates Wnt/β-catenin signaling via PI3K/Akt pathway [139]. Aberrant Wnt/β-catenin signaling drives SCD-induced fatty acid metabolism which forms a Wnt positive feedback loop via stabilization of Lrp5/6 [124]. FGF15, a mouse homologue of human FGF19, is known as the gut-derived hormone, which was reported to play role in modulating the metabolism in bile acid and carbohydrate [140]. A recent study showed that FGF15/FGFR4 signaling promoted activation of EMT and Wnt/β-catenin signaling cascade in the lipid metabolic disorder microenvironment [141]. 

## 5. The Role of β-Catenin in Cancer Stemness and Drug Resistance in HCC

### 5.1. The Role of Wnt/β-Catenin in the Regulation of Cancer Stemness

The Wnt/β-catenin signaling pathway plays an important role in stem cell biology, by regulating stem cell proliferation and oncogenesis, in which cancer stem cells are responsible for driving tumor heterogeneity [142]. Cancer stem cells (CSCs) exhibit features similar to those of normal stem cells, such as their self-renewal property. However, CSCs are more likely to develop tumor heterogeneity, which is not mediated by genetic mutations or epigenetic alterations, as previously discussed [143,144]. Moreover, β-catenin is known as a critical protein that is overexpressed in CSCs in many tumor tissues. The aberrant Wnt/β-catenin signaling pathway enhances CSC properties and allows the promotion of cancer growth [145,146]. To identify CSCs, examination of cell surface markers is one of the known methods, in which many markers modulate Wnt target genes, namely, leucine-rich repeat-containing G protein-coupled receptor 5/G protein-coupled receptor 49 (LGR5/GPR49), CD133, CD24, CD44, OV6 and epithelial cell adhesion molecule (EpCAM) [147]. Consistent evidence has shown that the expression of CSC markers is correlated with the activation of the canonical Wnt/β-catenin signaling pathway, which exhibited the significance of CSCs in enhancing the progression of HCC cells and leading to stemness mediated by β-catenin [19]. Lgr5 has been reported to be upregulated in various cancers, including HCC and breast and colorectal tumors [148,149,150]. It binds to the R-spongin ligand in Wnt signaling and stimulates stem cell properties [146]. Granulin-epithelin precursor (GEP), an oncofetal protein, was found to regulate liver CSCs and correlated with tumor recurrence [151]. Interestingly, GEP was regarded as the CSC marker in HCC, and was significantly correlated with β-catenin expression. EpCAM, another well-established liver CSC marker, was found to regulate cancer stemness and drug resistance via β-catenin/TCF1 positive feedback loop [152]. Notably, the ROS level is also associated with CSCs. The increase in GLS1 leads to a reduction of the ROS level, which enhances the transcription of Wnt target genes driven by β-catenin and enables an increase expression levels of certain stem cell markers, including CD44, Sox2 and Oct4, resulting in cancer stemness [71]. Conversely, high levels of ROS diminish stemness as Wnt signaling is hampered and CSC properties are decreased [153]. Another experiment showed that aquaporin-9 (AQP9) is weakly expressed in CSCs as it could reduce stemness by generating high levels of ROS, suppressing Wnt/β-catenin signaling and further leading to a reduction in CSC features [153]. In addition, protein tyrosine kinase 2 (PTK2) has been reported as tumor-promoting factors in HCC with their abilities to upregulate Wet signaling by maintaining CSC properties in HCC. PTK2 not only increases HCC cell viability and survival, but also induces resistance to sorafenib, enhancing CSC properties through the activation of Wnt/β-catenin signaling and initiating nuclear translocation of β-catenin [154]. Meanwhile, tumor-associated macrophages (TAM) type 2 that infiltrate the tumor microenvironment (TME) promote tumor necrosis factor-α (TNF-α) and further induce EMT in HCC [154,155]. It has been shown that the expression level of β-catenin is upregulated by M2 macrophages-derived TNF-α. Additionally, the risk of recurrence is predicted with increasing stemness as β-catenin is induced, suggesting that β-catenin and CSCs are positively correlated [155]. These results proved that CSCs are correlated with Wnt signaling, which is responsible for tumor recurrence and cancer stemness.

### 5.2. The Role of Wnt/β-Catenin in Drug Resistance

While it is known that drug resistance has been a persistent issue in treating cancers, it is possible to overcome drug resistance by targeting CSCs. Among a number of stem cell related pathways, alternation of Wnt/β-catenin signaling in HCC was shown to play a critical role in drug resistance to sorafenib, lenvatinib, cabozantinib and regorafenib, etc. [126]. A recent study showed that Interferon Regulatory Factor 2 (IRF2) regulated lenvatinib resistance in HCC cells via modulating the activity of β-catenin [156]. Another study revealed that regorafenib resistance occurs via the Wnt3a induced aberrant Wnt/β-catenin signaling. However, with the prolonged treatment of regorafenib, the resistant cells exhibited a diminishing Wnt/β-catenin activity with increased expression of liver CSC markers like CD24 and CD133 [157]. Our team has previously found that high levels of EPHB2 correlate with enhanced liver CSC properties and the regulation of sorafenib resistance via the SRC/AKT/GSK3β/β-catenin axis [158]. Additionally, a high level of cripto-1 sustained in either sorafenib-resistant HCC cells or patient-derived xenografts (PDXs) activates Wnt/β-catenin signaling by binding to FZD7/LRP6 and DVL3. On the other hand, downregulation of cripto-1 can reduce the crucial components of the Wnt pathway and suppress cancer stemness [159]. Consistently, our group have shown that UBE2T is involved in regulating liver CSCs by preventing β-catenin degradation via interacting with Mule expression via ubiquitination, suggesting that UBE2T is a regulator in drug resistance via the Wnt/β-catenin signaling pathway [160]. Other studies also revealed that GSK3β is a key player in modulating Wnt/β-catenin signaling through its function in regulating the degradation of β-catenin proteins [19]. SHP2 is a molecule that has been reported to be highly expressed in chemoresistant and recurrent HCC patients. It enhanced the nuclear translocation of β-catenin, facilitating β-catenin accumulation through the phosphorylation of GSK3β [161], in which this process allows the facilitation of self-renewal capabilities by CSC. Furthermore, Akt expression is enhanced in the presence of secretory clusterin (sCLu), promoting HCC development. The phosphorylation of GSK3β is then induced by Akt and leading to proteasomal degradation of β-catenin. Subsequently, liver CSC properties are maintained and drive sorafenib resistance in HCC [162]. Targeting signaling molecules crucial in Wnt/β-catenin potentially hampers cancer stemness and drug resistance in HCC (Figure 3).

As mentioned in the cancer metabolism section, there are several key enzymes and corresponding metabolites involved in glycolysis that play roles in the regulation of drug resistance. Enzymes such as hexokinase (HK), phosphofructokinase (PFK), and pyruvate kinase (PK), or their metabolites such as pyruvate and lactate, play roles in glycolytic-induced drug resistance through their antiapoptotic effects [91]. Upregulation of either of the mentioned elements enhanced glycolytic metabolism in tumor cells. Several studies have supported the assertion that the accumulation of either the HK or PFK isoform promotes chemoresistance and hinders apoptosis [163,164,165]. However, the M2 isoform of PK can shift to anaerobic glycolysis and induce the production of pyruvate and lactate in cancer cells Another study showed that knockdown of PKM2 can suppress the Warburg effect and overcome chemoresistance to cisplatin and doxorubicin [166,167]. In addition, the pentose phosphate pathway (PPP) branched from glycolysis is involved in producing R5P and the antioxidant nicotinamide adenine dinucleotide phosphate (NADPH) to synthesize nucleotides and maintain redox homeostasis between glutathione and thioredoxin, respectively [168]. Elevated PPP flux then elevates the level of NADPH, which enables cancer cells to survive oxidative stress induced by chemotherapy drugs, leading to evasion of apoptosis [169]. Apart from glycolytic enzymes and their metabolites, glucose transporters, mainly glucose transporter 1 (GLUT1) and glucose transporter 3 (GLUT3), regulate EMT-induced drug resistance [170]. Autophagy induction is another cause of drug resistance. Mammalian target of rapamycin (mTOR), the negative regulator of this self-digestion process, interacts with HK2, which leads to resistance to tamoxifen [171]. Furthermore, for the drug influx and efflux controlled by the pH gradient and ATP-binding cassette (ABC) transporters on the plasma membrane, extracellular acidification reduces the penetration and cytotoxicity of antitumor drugs, where the activities of ABC transporters are dependent on the levels of ATP generated via glycolysis in tumor cells [172,173]. In the condition of limited intracellular ATP levels, the function of ATP transporters is then inactivated, causing a failure in drug efflux and enhancing drug sensitivity [173]. 

## 6. Therapeutic Implications of Targeting the Wnt/β-Catenin Pathway in HCC

Wnt/β-catenin signaling plays multiple roles, which enables multifarious therapeutic strategies to target it. Recently, several drugs and molecules have been designed to target the Wnt/β-catenin cascade in HCC (Table 2). Several therapeutic approaches that involve using Wnt ligand inhibitors, targeting the extracellular components FZD/LRP, or targeting cytoplasmic molecules could be choices to inhibit Wnt signaling. Another option includes impeding nuclear transduction [174,175]. However, as of now, no clinically approved drugs can target this pathway effectively due to the drawbacks discovered within clinical trials in terms of safety and depressor effects [176]. 

### 6.1. Targeting the Interaction between Wnt Ligands and Their Receptors

There are several monoclonal antibodies that neutralize extracellular targets, including Wnt/FZD ligands and receptors, which play therapeutic roles. Antibodies against Wnt-1 and Wnt-2 have been raised to inhibit Wnt signaling and induce programmed cell death in HCC cell lines [181]. Apart from monoclonal antibodies, FzD receptors are another means to quench Wnt molecules. Recently, soluble FzD-7 (sFZD7) alone or in combination with doxorubicin has been shown to suppress Wnt signaling through the blockade of interactions between FZD/DVL, which decreasing the expression of Wnt target genes in HCC cells [207]. SFRP1 and WIF1 are Wnt antagonists that can block Wnt/β-catenin signaling [206], whereas IC-2 is a novel Wnt inhibitor that can reduce CD44+ populations and the formation of spheres in HCC [191]. Furthermore, a phase I trial with OMP-54F28, also called ipafricept (NCT02069145), showed that it acts as a recombinant fusion protein which can bind to Wnt ligands and antagonize the FZD8 receptor, leading to the suppression of Wnt-modulated processes. Moreover, a study targeting Wnt in HCC used DKN-01 as a combination therapy with gemcitabine and cisplatin in a phase I trial and with sorafenib in a phase II trial (NCT03645980). DKN-01 is a humanized monoclonal antibody against Dickkopf-1 (DKK1) that assists in hampering Wnt signaling by binding to the coreceptor LRP5/6. However, several studies may suggest conflicting or differing evidence regarding the oncogenic effects of DKK1 inhibition. [180,209,210,211]. Anti-Wnt2 and OMP-18R5 are the developed monoclonal antibodies preclinically targeting the binding of Wnt to Frizzled receptors [182,200].

### 6.2. Targeting the β-Catenin Destruction Complex

The presence of B-catenin destruction complex in the cytosol may serve as a therapeutic target for various cancers. Although it has not yet been further assessed in HCC, there are still some potentially significant reported conclusions. CGK062 is an antagonist that enhances proteasomal degradation of β-catenin by inducing β-catenin phosphorylation [185]. Whereas miRNA-1246 target both GSK3β and AXIN2 in regulating the Wnt signaling [197]. The application of TNKSi may likewise block Wnt signaling due to their ability to stabilize AXIN molecules and degrade β-catenin [212]. NVP-TNKS656 has been reported to inhibit HCC cell proliferation and suppress tumor growth in HCC preclinical models by attaching to several active pockets in tankyrase [198,199,213]. Furthermore, XAV939 also acts as a tankyrase inhibitor both in vitro and in vivo. A study showed that it could hamper the formation of colonies and proliferation of cells in HepG2 and Huh-7 HCC cell lines and inhibit HCC growth in xenograft models [214]. Pyrvinium represents another class of compounds that helps modulate the destruction complex. It binds to CK1α, thus activating it and subsequently increasing GSK3β expression for β-catenin phosphorylation [205].

### 6.3. Targeting β-Catenin/LEF-TCF Signaling

Various studies have shown the importance of forming β-catenin/LEF-TCF complexes in the nucleus, as this leads to downstream gene activation in promoting cancers. Therefore, the use of therapeutics targeting this binding could be a possible approach to treat cancers caused by the aberrant Wnt/β-catenin signaling pathway. Sorafenib is a multikinase inhibitor approved by the FDA that can suppress canonical Wnt signaling and can be used to target advanced-stage HCC. It can suppress TCF/LEF and the levels of β-catenin protein and Wnt-target gene mRNA [208]. However, there are reported cases of resistance with the upregulation of β-catenin. There are some Wnt/TCF inhibitors, including the fungal derivatives PKF115–854 and CGP049090, which play roles in blocking the interactions between β-catenin and TCF/LEF and reducing Wnt target gene expression [186,187], where ICG-001 blocks β-catenin/CREB binding [192,215]. It was shown that an ICG-001 inhibitor combined with sorafenib was used in a preclinical study, resulting in a further inhibitory effect against β-catenin [216,217]. Another combined treatment using FH-535 and sorafenib caused a significant drop in autophagic flux and enhanced apoptosis [218]. It is also noted that FH-535 decreased CD24 and CD44 expression in liver CSCs, thereby stimulating CSC properties and regulating β-catenin activities [176,219]. ISG12a was recently identified as an immune effector that hampers Wnt/β-catenin signaling through the inhibition of proteasomal degradation, further repressing the expression of PD-L1 and sensitizing HCC cells [220]. Porcupine (PORCN) is a membrane-bound O-acyltransferase that has been found to be useful in inhibiting the secretion of Wnt ligands by acylation [221,222]. PORCN inhibitors have been established to block the acylation of Wnt to prevent their secretion. There are clinical trials with either CGX1321 alone (NCT03507998) or combined with pembrolizumab (NCT02675946) in treating HCC patients. Consistently, treatment with CGX1321 resulted in prolonged survival and suppressed tumor burden in a mouse model. Furthermore, a psychiatric drug approved by the FDA, pimozide (PMZ), has been reported to inhibit HCC proliferation by lowering the expression of EpCAM downstream and disrupting the canonical Wnt pathway [203].

### 6.4. Targeting Other Components in the Wnt/β-Catenin Signaling Pathway

The Dishevelled protein was reported to be an upstream component in the Wnt/β-catenin signaling, which was found to play an important role in inducing Wnt/Fzd signaling. Targeting DVL by inhibiting DVL/FZD interaction via a KTXXXW-containing peptide was reported to inactivate the canonical Wnt/β-catenin pathway, leading to cell death of HCC cells. This result shows the blockade of DVL to FZD as a possible way to hamper the Wnt/β-catenin signaling [223]. Interestingly, there is a rise in the utilization of traditional Chinese medicines to inhibit HCC cell proliferation and induce cell cycle arrest through the blockade of Wnt/β-catenin signaling, such as agkihpin [224] Dendrobium candidum extract [225] and Zanthoxylum avicennae [226]. Curcumin is also a natural agent reported to have antitumor effects in HCC by modulating Wnt signaling [188]. 

## 7. Conclusions and Future Perspectives

Since the Wnt/β-catenin signaling pathway was identified decades ago, substantial attention has been drawn to the field of HCC research due to its critical involvement in tumor initiation, progression and therapeutic resistance. The present review highlighted the role of Wnt/β-catenin in the regulation of cancer stemness and metabolic reprogramming, which are regarded as therapeutic vulnerabilities against HCC. However, our understanding of how the Wnt/β-catenin pathway contributes to these cancer hallmarks remains insufficient. Therefore, additional molecular and animal studies are required urgently to identify novel therapeutic targets against the Wnt/β-catenin pathway for HCC treatment. Recent pre-clinical study showed that Wnt/β-catenin is one of the major pathways hindering the efficiency of current immune checkpoint therapy [227]. Echo to this finding, alternations of Wnt/β-catenin pathway was correlated with lower disease control rate, shorter median disease-free and overall survivals of HCC patients [228]. With the encouraging results of combination of atezolizumab and bevacizumab in advanced HCC patients [8], targeting Wnt/β-catenin pathway in combination with immune checkpoint therapy may obtain a good therapeutic outcome. In HCC, only a few molecular targeted drugs, such as sorafenib, lenvatinib and regorafenib, have been FDA-approved for the treatment of HCC. Although these drugs offer great promise for targeting the Wnt/β-catenin pathway to treat HCC patients, the clinical efficacy of targeting these pathways remains uncertain and is still undergoing clinical trials.

## Figures and Tables

**Figure 1 cancers-14-05468-f001:**
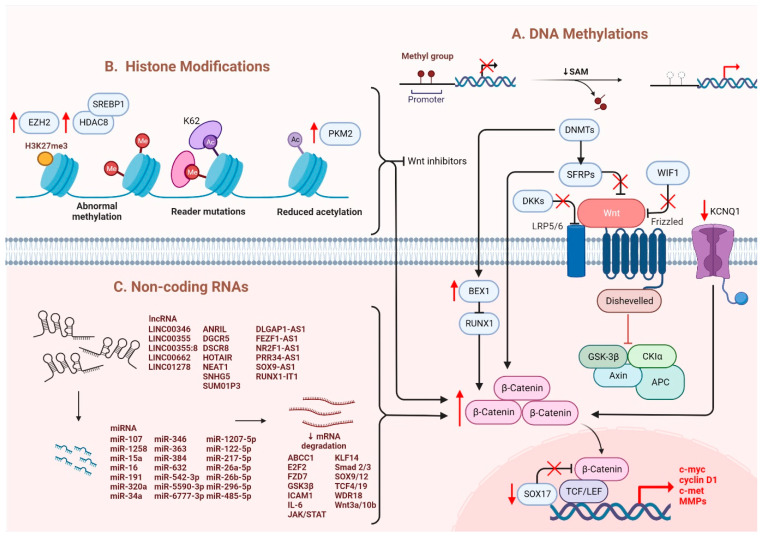
Regulation of Wnt/β-catenin signaling in HCC. Wnt/β-catenin signaling in HCC is regulated by (**A**) DNA methylation, (**B**) histone modification and (**C**) non-coding RNAs.

**Figure 2 cancers-14-05468-f002:**
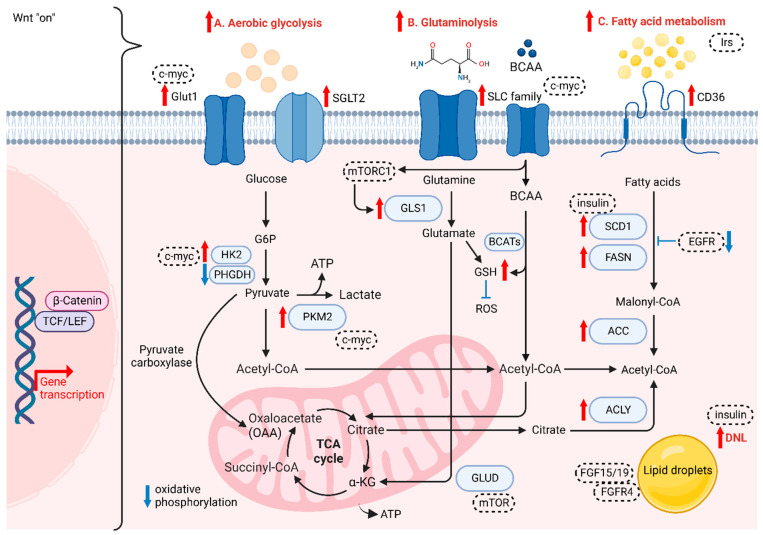
The role of Wnt/β-catenin signaling on cancer metabolism. Wnt/β-catenin plays crucial role in (**A**) aerobic glycolysis, (**B**) glutaminolysis and (**C**) fatty acid metabolism of HCC cells.

**Figure 3 cancers-14-05468-f003:**
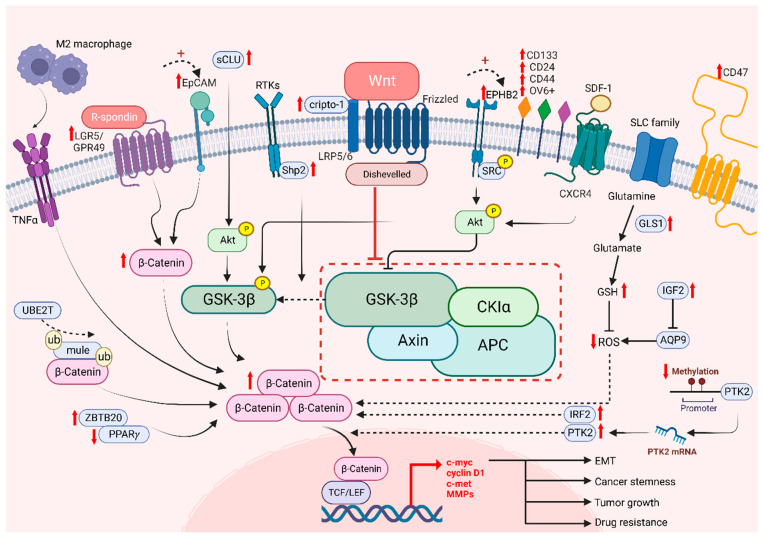
The role of Wnt/β-catenin signaling on cancer stemness and drug resistance. Intrinsic and extrinsic regulation of Wnt/β-catenin signaling lead to tumor growth, cancer stemness, EMT and drug resistance of HCC cells.

**Table 1 cancers-14-05468-t001:** The list of lncRNAs and their related miRNAs in regulation of Wnt/β-catenin signaling in HCC.

LncRNA	miRNA	Targets	Ref.
LINC00346	miR-542-3p	FZD7, WDR18	[54]
LINC00355	miR-217-5p	GSK3β, c-myc, CCND1	[46]
LINC00355:8	miR-6777-3p	Wnt10b	[55]
LINC00662	miR-15a, miR-16, miR-107	Wnt3a	[48]
LINC01278	miR-1258	TCF-4, Smad2/3	[47]
ANRIL	miR-191, miR-122-5p	CCND1, p53, p21, MMP-2, MMP-9, Vimentin	[56,57]
DGCR5	miR-346	KLF14	[58]
DSCR8	miR-485-5p	FZD7	[59]
DLGAP1-AS1	miR-26a-5p, miR-26b-5p	IL-6, JAK2, STAT3	[60]
FEZF1-AS1	miR-107	Wnt3a, ICAM1, Vimentin	[49]
HOTAIR	miR-34a	Akt	[61]
MIR194-2HG	miR-1207-5p	TCF19	[62]
NEAT1	miR-384	Wnt	[63]
NR2F1-AS1	miR-363	ABCC1	[64]
PRR34-AS1	miR-296-5p	E2F2, SOX12	[65]
RUNX1-IT1	miR-632	GSK3β	[66]
SNHG5	miR-26a-5p	GSK3β	[67]
SUMO1P3	miR-320a	C-myc, CCND1	[68]
SOX9-AS1	miR-5590-3p	SOX9	[69]

**Table 2 cancers-14-05468-t002:** Preclinical and clinical studies on targeting Wnt/β-catenin signaling pathway in HCC.

Clinical				
Agents	Targets/Mechanisms	Phase	Clinical Trial no.	Ref.
CGX1321	PORCN inhibitor	I	NCT03507998	[177]
CGX1321 (With pembrolizumab)	PORCN inhibitor	I	NCT02675946	[178]
OMP-54F28 (with sorafenib)	FZD8 decoy receptor	I	NCT02069145	[179]
DKN-01	DDK1	I/II	NCT03645980	[180]
**Preclinical**				
Agents/Molecules	Roles/Mechanisms			Ref.
Anti-Wnt1 mAb	Inhibit Wnt signaling and Wnt related ligands			[181]
Anti-Wnt2 mAb	Blockade of Wnt binding to FZD receptors Inhibit Wnt signaling and Wnt related ligands			[181,182]
Bcl9/9L	Inactivate Wnt signaling			[183]
BrMC	Inhibit CD133+ liver CSCs proliferation Decrease β-catenin expression			[184]
CGK062	Enhance proteasomal degradation of β-catenin Induce β-catenin phosphorylation			[185]
CGP049090	Blockade of β-catenin/TCF/LEF interaction Decrease Wnt-target genes expression			[186,187]
*Curcumin*	Inhibit GPC3/TPA-induced Wnt activation			[188]
Epigallocatechin-3-gallate	Inhibit Wnt via suppressing c-myc while activating SFRP1			[189]
FH-535	Inhibit the activation of β-catenin regulated genes			[176]
FH-535 (Combined with sorafenib)	Inhibit Wnt-target genes Suppress the recruitment of β-catenin			[190]
IC-2	Wnt/β-catenin signaling			[191]
ICG-001	Disrupt the binding of β-catenin-CREB			[192]
LncRNA-RUNX1	Increase GSK3β expression for β-catenin phosphorylation			[193]
LncRNA-Mir22HG	Inactivate β-catenin via downregulation of miRNA-10a-5p			[194]
miRNA-885-5p	Decrease CTNNB1 expression level			[195]
miRNA-122	Decrease MDR1 expression level and inhibit Wnt/β-catenin			[196]
miRNA-1246	Target the Wnt/β-catenin degradation complex, GSK-3β and AXIN2			[197]
NVP-TNKS656	Tankyrase inhibitor			[198,199]
OMP-18R5	Blockade of Wnt binding to FZD receptors			[182,200]
Ornithine aminotransferase (OAT)	Inactivators Decrease L-Gln and AFP serum levels			[201]
PKF115-854	Blockade of β-catenin/TCF/LEF interaction Decrease Wnt-target genes expression			[186,187]
PKF118-310	Blockade of β-catenin/TCF/LEF interaction Decrease Wnt-target genes expression			[202]
Pimozide	Antipsychotic drug Inhibit the degradation complex and Wnt signaling			[203]
PMED-1	Blockade of β-catenin/CBP interaction			[204]
Pyrvinium pamoate	Increase GSK3b expression for β-catenin phosphorylation			[205]
SFRP1	Wnt antagonists Blockade of Wnt signaling			[206]
sFZD7 (Or combined with doxorubicin)	Blockade of FZD/DVL interaction Decrease Wnt-target genes expression			[207]
Sorafenib	Decrease TCF/LEF, β-catenin and Wnt-target genes mRNA levels			[208]
WIF1	Wnt antagonists Blockade of Wnt signaling			[206]

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
