# Peer review of "Wnt/β-Catenin Signaling as a Driver of Stemness and Metabolic Reprogramming in Hepatocellular Carcinoma"

_cancers, 2022, doi:10.3390/cancers14215468_

Round 1

Reviewer 1 Report

The present review article written by LEUNG and LEE titled “ Wnt/β-catenin signaling as a driver of stemness and metabolic reprogramming in hepatocellular carcinoma” reviewed the role of Wnt/β-catenin signaling in stemness and metabolic reprogramming mediated development of hepatocellular carcinoma. The manuscript is well written and drafted nicely. The abstract is well written and conclusive. Authors listed the therapeutic/clinical trial inhibitors and described well. The figures are very informative and constructed impressively. Authors can summarize the figure briefly in the respective figure legend sections.

Author Response

Reviewer #1

The present review article written by LEUNG and LEE titled “ Wnt/β-catenin signaling as a driver of stemness and metabolic reprogramming in hepatocellular carcinoma” reviewed the role of Wnt/β-catenin signaling in stemness and metabolic reprogramming mediated development of hepatocellular carcinoma. The manuscript is well written and drafted nicely. The abstract is well written and conclusive. Authors listed the therapeutic/clinical trial inhibitors and described well. The figures are very informative and constructed impressively. Authors can summarize the figure briefly in the respective figure legend sections.

Response: Thank you very much for your comments to this review article.

Reviewer 2 Report

This article provides a comprehensive overview of the Wnt/b-catenin signaling, which has been proven to be important for the development, growth, and clinical impact of HCC. The relevance of Wnt/b-catenin signaling in developing cancer cell survival and metabolic remodeling, which are regarded as emerging cancer hallmarks, has been highlighted by mounting data.  The regulatory functions of Wnt/b-catenin signaling in metabolic transformation, cancer cell survival, and drug resistance in HCC also were updated by Leung et al. In addition, this review sheds light on the potential therapeutic targets of the Wnt/b-Catenin signaling pathway.

1-Previous reviews that addressed the function of Wnt-Hippo signaling pathways involved in hepatocellular carcinoma also need to be cited in this manuscript. 

doi: 10.3390/cancers14194580

doi: 10.1038/s41392-021-00701-5

DOI: 10.2174/187152011796011019

DOI https://doi.org/10.2147/JHC.S336858

2- The Wnt/b Catenin pathway is also regulated by reactive oxygen species, which also needs elaboration in the text.

3-Please also discuss the role of the Wnt signaling pathway in branched-chain amino acid metabolism in liver cells.

4-Wnt/b-catenin signaling modifies the activity of c-myc, AMPK, and mToR, which are known to regulate cellular metabolism. All of these relationships must be added in the text and illustrations in detail.

5- Additionally, how do hormones and growth factors (such as insulin, EGF, FGF, and TGF) influence cellular metabolism and b-Catenin activity? should be addressed in the text and images.

Author Response

This article provides a comprehensive overview of the Wnt/b-catenin signaling, which has been proven to be important for the development, growth, and clinical impact of HCC. The relevance of Wnt/b-catenin signaling in developing cancer cell survival and metabolic remodeling, which are regarded as emerging cancer hallmarks, has been highlighted by mounting data.  The regulatory functions of Wnt/b-catenin signaling in metabolic transformation, cancer cell survival, and drug resistance in HCC also were updated by Leung et al. In addition, this review sheds light on the potential therapeutic targets of the Wnt/b-Catenin signaling pathway.

1-Previous reviews that addressed the function of Wnt-Hippo signaling pathways involved in hepatocellular carcinoma also need to be cited in this manuscript. 

doi: 10.3390/cancers14194580

doi: 10.1038/s41392-021-00701-5

DOI: 10.2174/187152011796011019

DOI https://doi.org/10.2147/JHC.S336858

Response: Thank you very much to your comments. We have incorporated the comments on page 11 and added references (67, 68).

2- The Wnt/b Catenin pathway is also regulated by reactive oxygen species, which also needs elaboration in the text.

Response: Thank you very much for your comments. We have incorporated the comments and added some paragraphs on page 14 under “ Other molecules involved in the regulation of Wnt/beta-catenin.”

3-Please also discuss the role of the Wnt signaling pathway in branched-chain amino acid metabolism in liver cells.

Response: Thank you very much for your comments. We have incorporated the comments and added some paragraphs on page 10 to 11 under “ The role of Wnt/β-catenin in glutaminolysis”.

4-Wnt/b-catenin signaling modifies the activity of c-myc, AMPK, and mToR, which are known to regulate cellular metabolism. All of these relationships must be added in the text and illustrations in detail.

Response: Thank you very much for your comments. We have incorporated the comments and added some paragraphs on page 16 under “The roles of growth factors and oncogenic factors in regulation of Wnt/β-catenin- induced cellular metabolism”

5- Additionally, how do hormones and growth factors (such as insulin, EGF, FGF, and TGF) influence cellular metabolism and b-Catenin activity? should be addressed in the text and images.

Response: Thank you very much for your comments. We have incorporated the comments and added some paragraphs on page 16 under “The roles of growth factors and oncogenic factors in regulation of Wnt/β-catenin- induced cellular metabolism”

Reviewer 3 Report

The authors reviewed the regulatory mechanism of Wnt/β-catenin signaling and its role in hepatocellular carcinoma (HCC). They also stated the regulatory roles of Wnt/β-catenin signaling in drug resistance in HCC. The review article is well written and interesting. On the other hand, I suggest some minor revisions to improve this work.

Minor revisions

1)     If the authors intended to argue the role of Wnt/β-catenin signaling in drug resistance, they should argue about the other tyrosine kinase inhibitors including lenvatinib,. cabozantinib, regorafenib.

2)     Considering the era in combination immunotherapy such as atezolizumab and bevacizumab for patients with HCC, it would be useful to argue the relationship between immune check point inhibitors and Wnt/β-catenin mutation.

Additional reference recommended: Harding JJ,et al. Prospective Genotyping of Hepatocellular Carcinoma: Clinical Implications of Next-Generation Sequencing for Matching Patients to Targeted and Immune Therapies. Clin Cancer Res. 2019 Apr 1;25(7):2116-2126. doi: 10.1158/1078-0432.

3)     Page 9 Line32: Please cite an appropriate paper.

Author Response

The authors reviewed the regulatory mechanism of Wnt/β-catenin signaling and its role in hepatocellular carcinoma (HCC). They also stated the regulatory roles of Wnt/β-catenin signaling in drug resistance in HCC. The review article is well written and interesting. On the other hand, I suggest some minor revisions to improve this work.

Minor revisions

If the authors intended to argue the role of Wnt/β-catenin signaling in drug resistance, they should argue about the other tyrosine kinase inhibitors including lenvatinib,. cabozantinib, regorafenib.

Response: Thank you very much for your comments. We have incorporated the comments and added some paragraphs on page 18 under “The role of Wnt/β-catenin in drug resistance”

Considering the era in combination immunotherapy such as atezolizumab and bevacizumab for patients with HCC, it would be useful to argue the relationship between immune check point inhibitors and Wnt/β-catenin mutation.

Response: Thank you very much for your comments. We have incorporated the comments and added some paragraphs on page 23 under “Conclusions and future perspectives”

Additional reference recommended: Harding JJ,et al. Prospective Genotyping of Hepatocellular Carcinoma: Clinical Implications of Next-Generation Sequencing for Matching Patients to Targeted and Immune Therapies. Clin Cancer Res. 2019 Apr 1;25(7):2116-2126. doi: 10.1158/1078-0432.

Response: Thank you very much for your suggestions. We have incorporated the comments and added this reference [198].

Page 9 Line32: Please cite an appropriate paper.

Response: Thank you very much for your comments. The appropriate paper has been inserted.

Round 2

Reviewer 2 Report

The authors have done a great job in this review. Compliments!